# Insecticidal activity of *Ageratina adenophora* (Asteraceae) extract against *Limax maximus* (Mollusca, Limacidae) at different developmental stages and its chemical constituent analysis

**Haojun Li**[ORCID]**, Runa Zhao, Yingna Pan, Hui Tian, Wenlong Chen** *

Provincial Key Laboratory for Agricultural Pest Management of Mountainous Region, Institute of Entomology, Guizhou University, Guiyang, China

* wlchen@gzu.edu.cn

## Abstract

*Limax maximus*, or great gray slug, is a common agriculture pest. The pest infests crops during their growth phase, creating holes in vegetable leaves, particularly in seedlings and tender leaves. A study was conducted to assess the insecticidal activity of *Ageratina adenophora* extract against these slugs. Factors such as fecundity, growth, hatching rate, offspring survival rate, protective enzyme activity, and detoxifying enzyme activity were examined in slugs exposed to the extract's sublethal concentration ($LC_{50}$) for two different durations (24 and 48 h). The phytochemical variability of the extracts was also studied. The $LC_{50}$ value of the *A. adenophora* extract against *L. maximus* was 35.9 mg/mL. This extract significantly reduced the hatching rate of eggs and the survival rate of offspring hatched from exposed eggs compared with the control. The lowest rates were observed in those exposed for 48 h. The survival, growth, protective enzyme, and detoxification activity of newly hatched and 40-day-old slugs decreased. The *A. adenophora* extract contained tannins, flavonoids, and saponins, possibly contributing to their biological effects. These results suggest that the extract could be used as an alternative treatment for slug extermination, effectively controlling this species.

## 1. Introduction

The terrestrial slug *Limax maximus* (Limacidae) is a significant agricultural pest in China owing to its widespread distribution and numerous host species. Both adult and larvae can cause damage to crops, especially those in the Solanaceae, Cruciferae, and Leguminosae families. They leave white mucus traces and sticky feces after crawling, which can harbor bacteria. The damage caused by slugs, including bite marks on leaves, fruits, and stems, significantly impacts the quality of agricultural products and crop safety [1]. Slugs not only harm crops but are also closely associated with plant pathogens and the transmission of parasites to humans, livestock, and wild mammals, posing a considerable threat to human health.

**Data Availability Statement:** Our data cannot be made public because of the restrictions of a third party, the Institute of Entomology of Guizhou

University. Our data are all owned by the College of Agriculture of Guizhou University, which is the superior unit of the Institute of Entomology of Guizhou University. If the data are provided as required, you can contact Zhou Wei (Zw18786131893@126.com), a researcher at the Institute of Entomology, Guizhou University, and Li Zhili (3159517137@qq.com), a researcher at the School of Agriculture, Guizhou University, who is responsible for data induction and storage.

**Funding:** The author(s) received no specific funding for this work.

**Competing interests:** The authors have declared that no competing interests exist.

Currently, the control of *L. maximus* primarily involves spraying chemical pesticides, such as tetrameric acetaldehyde [2]. However, the prolonged use of slug control agents can lead to slug resistance and diminish the control efficacy. Additionally, these substances can harm the environment, especially by causing water pollution [3]. When in danger, the epidermal glands of *L. maximus* secrete a large amount of mucus, rendering it difficult for drugs to act directly on *L. maximus* [4]. For *L. maximus*, metaldehyde is a toxic pesticide with a long half-life; hence, it is very easy to form pesticide residues and harm numerous nontarget organisms [5]. Some people also catch slugs with their hands. However, *L. maximus* is generally active at night and manual cleaning is time-consuming and costly. In addition, *L. maximus* carries parasites; thus, there is a risk of parasite infection if protective measures are not taken during cleaning. Lime is generally used to control *L. maximus*; however, it is difficult to administer in large-scale farms [6]. At present, to prevent and biologically control *L. maximus*, *Carabus elysii* is used to develop new low-toxicity pesticides; however, *C. elysii* is difficult to raise and release at a fixed point on a large scale [1]. Therefore, it is an important trend to study pesticides with low preparation cost and toxicity.*Ageratina adenophora* (Asteraceae) is a perennial herb of the Compositae family, native to South America [7]. It was introduced into Lincang, Yunnan, China, from Myanmar in the 1940s and has become widely distributed in Yunnan, Guizhou, Sichuan, Guangxi, and Chongqing. In 2003, it was listed as an alien invasive species in China. This plant often thrives in tidal wetlands or on hillside roads, exhibiting robust fruiting ability and rapid spread. *A. adenophora* has strong vitality and fecundity, displaying broad adaptability [8]. It invades farmlands, gardens, forests, and other habitats, forming dominant communities, suppressing the growth of other plants, and disrupting plant diversity, especially in southwest China [6]. Studies have shown that this plant contains active principles in its chemical constituents, such as flavonoids, saponins, and tannins, possessing biocidal activities, including molluscicidal properties [1, 7–9]. These findings provide a feasibility for this plant to control *L. maximus*.

This study aimed to confirm the presence of flavonoids, tannins, and saponins in the extract of *A. adenophora* and to analyze and identify their contents in the extract. We calculated the effects of $LC_{50}$ on eggs, larvae, and adult slugs and investigated the impact of 24 and 48 h exposure to semi-lethal concentrations on different life stages of slugs.

## 2. Materials and methods

### 2.1 Slugs and plants

The eggs, newly hatched slugs, and 40-day-old slugs utilized in this study were sourced from the ecological experimental field of Professor Chen at the Institute of Entomology, Guizhou University (latitude: 26°25′39.62″N; longitude: 106°40′5.81″E; 1090 m altitude). Slugs were reared following a standard method [10]. In summary, *L. maximus* were reared in plastic containers with dimensions of 9 cm in diameter and 6 cm in length, containing 50 g of damp culture soil (120° 1 h), and a mixture of fresh vegetables and protein-rich feed mass (3:1) for feeding. The top of the container was covered with a plastic film with small holes for ventilation. Environmental conditions were maintained at 25 ± 1°C, 70 ± 5% relative humidity, and a 16:8 h (L:D) photoperiod in an environmental chamber. The *L. maximus* selected in this study are shown in Fig 1.

Leaves of *A. adenophora* were collected from the campuses of Guizhou University (latitude: 26°42′69.93″N; longitude: 106°66′95.34″E; 1083 m altitude) in Guiyang in December 2022. These leaves were cleaned, dried in an oven at 60°C for 24 h, and ground into a powder using a mechanical grinder (Qingdao Juchuang Environmental Protection Group Co., Ltd. JC-FW-100). The plant extract was prepared by infusing 100 g of leaf powder in 1000 mL of distilled

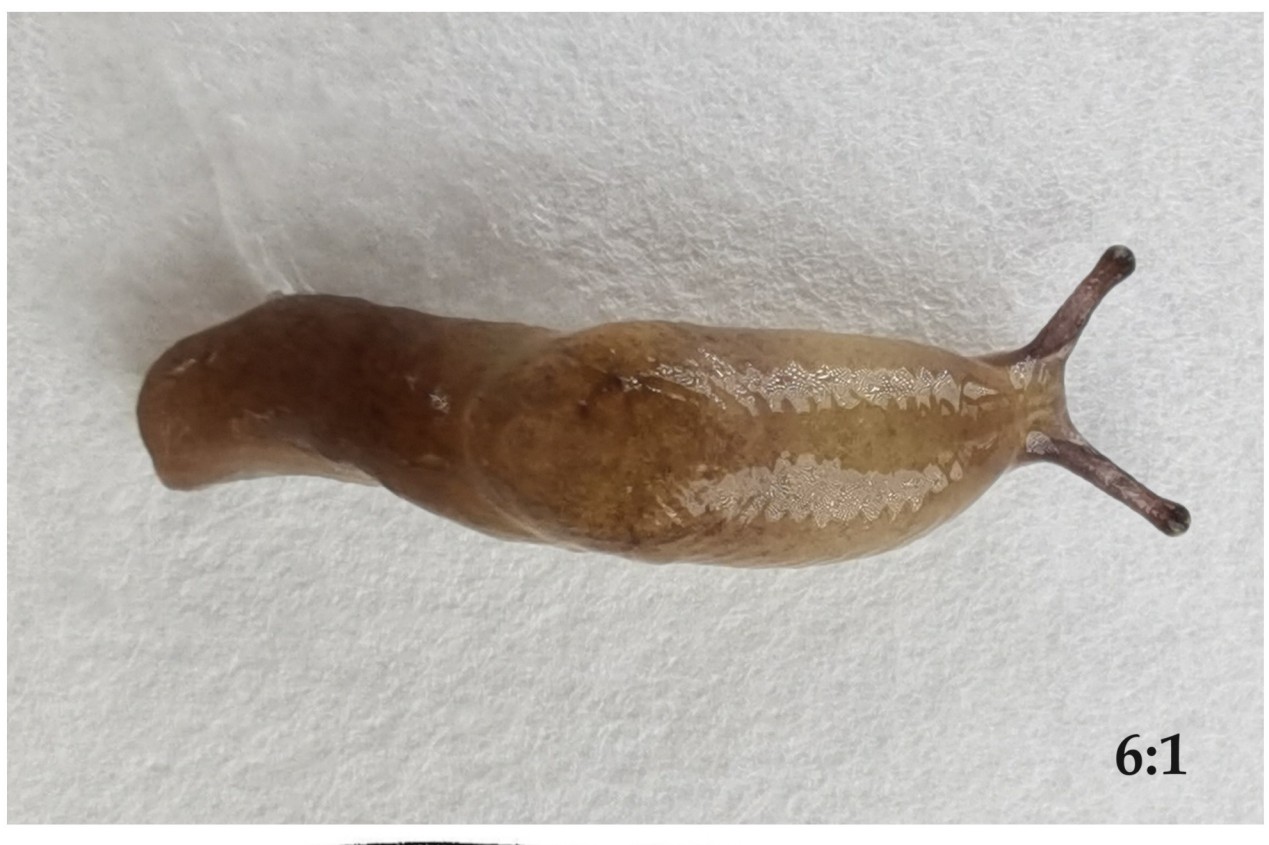

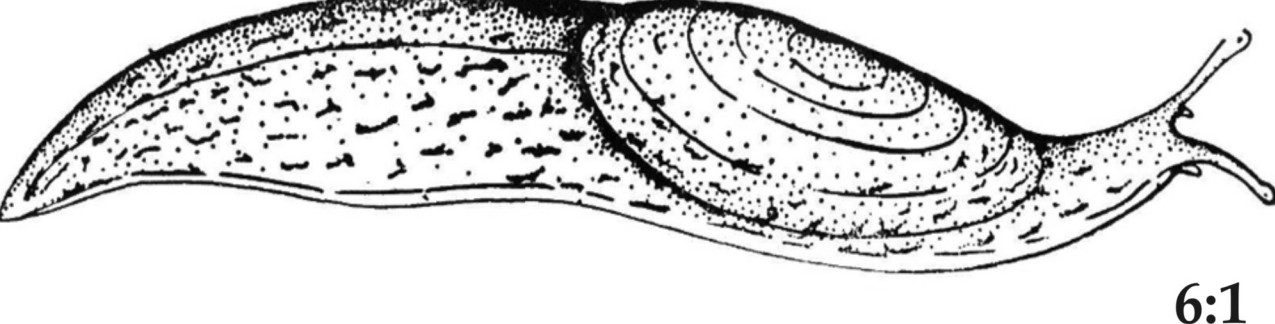

**Fig 1. Morphological schematic diagram of *L. maximus*.**

water for 72 h, then filtering the obtained suspension and testing it with an aqueous solution. The sublethal concentration ($LC_{50}$) for adult slugs was calculated for subsequent hatching and survival experiments. The *A. adenophora* selected in this study are shown in Fig 2.

This study and included experimental procedures were approved by the institute of entomology, Guizhou University. All animal housing and experiments were conducted in strict accordance with the institute crawn id sines for care and use of laboratory animals.

## 2.2 Phytochemical analysis of aqueous extract of *A. adenophora*

To determine the presence of various compounds such as flavonoids, saponins, and tannins, 5.0 g of dried powder was boiled in distilled water for 10 min. The solution was then filtered

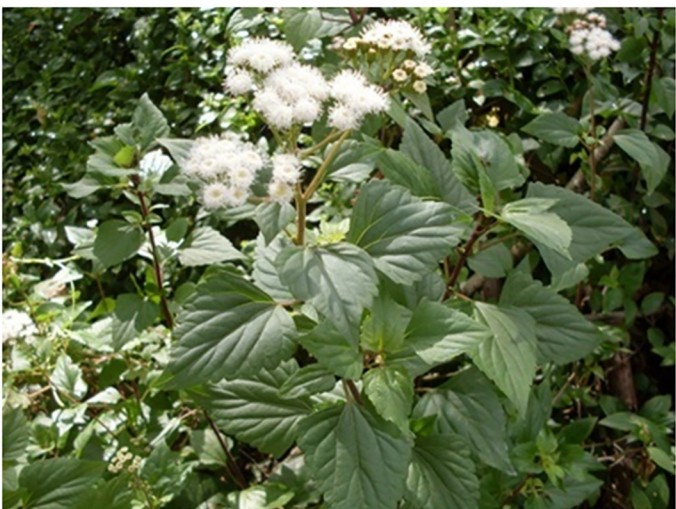

**Fig 2. Morphological schematic diagram of *A. Adenophora*.**

after cooling, and the supernatant was used for detection. The presence of flavonoids was confirmed using a 10% sodium hydroxide colorimetric analysis [11]. The identification of saponins and the determination of the foaming index in the water extract were performed according to the standard method [12, 13]. To identify tannins in the *A. adenophora* extract, a 2.5% agar solution was prepared in distilled water. Colorimetric tests were performed with a 2% ferric chloride ($FeCl_3$) solution to distinguish between types of tannins (condensed or hydrolyzed). Condensed tannins were quantified using the Stiasny method [14], which was repeated three times, and the presence of tannins was confirmed based on the turbidity of the water extract.

## 2.3 Determination of flavonoids, saponins, and tannins in *A. adenophora* aqueous extract

The sample solution was prepared with 1000 mL of the extract. The flavonoid content was assessed using a colorimetric method [15], and rutin was utilized as a standard control. Briefly, 2 mg of rutin was dissolved in ethanol via ultrasonication to create a standard solution of 200 μg mL$^{-1}$. Volumes of 0, 0.1, 0.2, 0.3, 0.3, 0.4, 0.5, and 0.6 mL were precisely measured and added to 2 mL of water, labeled as 1–7, and 100 μL respectively. The corresponding reagent without rutin served as the blank reference. The absorbance (OD) value was read at 410 nm using a full-wavelength microplate reader, and the standard curve was calculated [16]. For the color operation, 1.0 mL of the sample solution was used, and the OD value was determined at a wavelength of 410 nm using a full-wavelength enzyme labeling instrument. This process was repeated six times, and the data were recorded and analyzed. The saponin content was determined using a modified method [17]. To 20 μL of the aqueous extract of *A. adenophora*, after the evaporation of ethanol, 5 mL of 72% sulfuric acid and 0.5 mL of 8% vanillin ethanol solution were added. The solution was gently mixed, and the OD was determined at 544 nm using a full-wavelength microplate reader. The standard curve of ginsenoside Re was employed to calculate the saponin content. The experiment was repeated six times. The total tannin content was estimated using a previously described procedure [18]. To 1 mL of the extract, 5.0 mL of water, 1.0 mL of sodium tungstate–sodium molybdate mixed solution, and 3.0 mL of sodium

carbonate solution were successively added for the color reaction. The OD value was determined at 765 nm using a full-wavelength microplate reader. The standard curve of gallic acid was used for determining the tannin content. The experiment was repeated six times for statistical analysis.

The contents of flavonoids, saponins, and tannins in *A. adenophora* extract were calculated [19] according to the following formula:

$$(\%) = \left( \frac{m \times V_2}{V_1 \times M \times 1000} \right) \times 100\% \tag{1}$$

Where, m is the mass of flavonoids, saponins, and tannins; $V_1$ is the sampling volume of the sample solution; $V_2$ is the constant volume of the solution; and M is the sample mass.

## 2.4 Analysis of hatching rate of exposed eggs

The $LC_{50}$ of the *A. adenophora* aqueous extract was utilized for bioassays against the eggs of great gray slugs. In each Petri plate (diameter 9 cm), 10 1-day-old eggs were placed. The eggs were then sprayed with 20 mL of the extract. For the control group, the eggs were treated with distilled water only. After 24 and 48 h, the eggs were transferred to plastic containers. The hatching of eggs was observed daily for 30 days. Each treatment was replicated four times. The experiment was conducted under 21˚C–24˚C and 84% relative air humidity. The hatching rate (%) was determined using the following formula [20]: hatching rate (%) = (number of slugs hatched/total number of eggs) × 100.

## 2.5 Survival rate of newly hatched offspring

The hatched slug larvae were maintained under the previously described conditions. They were provided a diet comprising fresh vegetable leaves and a protein-rich feed mixture (3:1). The incubator was moistened with distilled water every 2 days. Mortality and maturity assessments were conducted every 3 days over 120 days, and any deceased individuals were promptly removed from the incubator. The survival rate (%) was calculated using the following formula [21]: Survival rate (%) = (number of unviable embryos/total number of embryos) × 100.

## 2.6 Effects of $LC_{50}$ on the growth, survival, and reproduction of newly hatched larvae and *L. maximus*

In this experiment, 60 newly hatched slug larvae and 40-day-old slugs were selected, exhibiting an average size of 6.62 ± 0.57 mm (for the newly hatched larvae) and 38.48 ± 2.14 mm (for the 40-day-old slugs). *L. maximus* were organized into groups (10 in each group, with four repetitions) and placed in plastic perforated boxes filled with presterilized culture soil. A precalculated $LC_{50}$ of *A. adenophora* extract (20 mL) was sprayed in each box. The control group, containing the same number of *L. maximus*, was treated with distilled water only and sealed with cotton cloth and elastic rubber to prevent slug escape. At the end of the exposure period, the slugs were transferred to another container and maintained using the method described above.

Growth was assessed by measuring body length using Vernier calipers [22]. The mortality rate was evaluated every 3 days over 120 days, and any deceased individuals were promptly removed from the incubator via direct observation of *L. maximus*. Sexual maturation was also assessed at the same frequency, determined by the presence of eggs in the womb [23].

## 2.7 Effects of $LC_{50}$ on the activities of protective and detoxifying enzymes in newly hatched larvae and slugs

Superoxide dismutase (SOD), peroxidase (POD), and catalase (CAT) constitute the protective enzyme system in mollusks. These enzymes work together to maintain intracellular free radicals at a low level, effectively preventing free radical toxicity. On the contrary, cytochrome P450 (CYP450), glutathione-S-transferase (GST), and acetylcholinesterase (AChE) are prominent detoxification enzymes in mollusks. These enzymes are important in breaking down exogenous toxins and maintaining normal physiological metabolism [24]. The average sizes of the newly hatched slug larvae and 40-day-old slugs were $6.62 \pm 0.57$ mm and $38.48 \pm 2.14$ mm, respectively. The slugs were placed in plastic boxes with holes and raised in the same environment as mentioned earlier. Healthy slugs were subjected to the precalculated $LC_{50}$. The control group, comprising the same number of slugs, was treated with distilled water only. The treatment duration was set at 24 and 48 h, and each was repeated five times within these periods. After the treatment, the slugs were preserved in liquid nitrogen and weighed postmortem. A 10% adult homogenate was prepared by adding a volume of homogenate medium (0.1 mol/L saline, pH 7–7.4) nine times the sample's weight (in grams) under an ice bath. The mixture was then centrifuged at 4000 rpm for 10 min. The supernatant was collected, and its OD value was determined at 420 nm with a 1 cm light path using a 100 μL sample.

Inhibition activities of protective enzymes, i.e., SOD, POD, and CAT, and detoxification enzymes, i.e., CYP450, GST, and AChE, were determined as per the commercial enzyme-linked immunosorbent assay kit (Suzhou Mengxi Biopharmaceutical Technology Co., Ltd.). The newly hatched slug larvae and adults treated with $LC_{50}$ of *A. adenophora* aqueous extract or distilled water for 24 and 48 h were collected for enzyme assay. The treated slugs were stored at −80°C.

## 2.8 Statistical analysis

The regression equations for the standard samples used to determine flavonoids, saponins, and tannins were calculated and analyzed using the Microsoft Excel 2021 software. The $LC_{50}$ value was computed using the Probit analysis in IBM SPSS Statistics 25 software [9]. The concentration–response curve was established using the same test [25]. To compare hatchability, mortality, growth, and development of *L. maximus* exposed to $LC_{50}$, the Kruskal–Wall is and the Student–Kewman–Keuls methods ($p < 0.05$) were employed. The hatching rate, mortality, and growth of slugs exposed to $LC_{50}$ were compared using the BioEstat 5.0 software. Statistical analysis of protective and detoxification enzyme activity data was performed using IBM SPSS Statistics 25. The difference between treatments was analyzed using single-factor analysis of variance, and the significant difference was tested using the least significant difference method. Curves were plotted using the Origin 2021 software.

# 3. Results

## 3.1 Phytochemical analysis of *A. adenophora* extract

After adding the sodium hydroxide solution, the *A. adenophora* extract changed from light yellow to red, a characteristic color associated with chalcone and gold ketone, thus confirming the presence of flavonoids in the aqueous extract. Tannin presence was confirmed using the agar solution test, with the emergence of a moss green color affirming the presence of condensed tannin upon adding ferric chloride. This finding was further validated using the Stiasny test. The results of these analyses unequivocally indicated the presence of saponins, as evidenced by a foam index of 100.

## 3.2 Determination of flavonoids, saponins, and tannins in the extract

**3.2.1 Calculation of standard curve of flavonoids, saponins and tannins.** The standard curve of flavonoids was determined using rutin. The regression equation for the rutin reference substance was expressed as y = 0.1744x + 0.0007, with an $R^2$ value of 0.9946. This signifies a robust linear relationship for the rutin reference substance within the specified range of 0.00–0.10 μg. The standard curve of saponins was determined by ginsenoside Re. The regression equation for the ginsenoside Re reference substance was described as y = 0.0427x + 0.0119, with an $R^2$ value of 0.9965. This indicates a strong linear relationship for the ginsenoside Re reference substance within the specified range of 0.00–10.00 μg. The standard curve of saponins was determined by gallic acid. The regression equation for the gallic acid reference substance was y = 0.2082x + 0.0038, with an $R^2$ value of 0.9968. This signifies a good linear relationship for the gallic acid reference substance within the specified range of 0.00–0.60 μg. Fig 3 represents the standard curve, plotting mass (in μg) on the X-axis and OD values on the Y axis.

**3.2.2 The content of the extract.** After conducting six repeated tests, the content of flavonoids in the *A. adenophora* extract ranged from 4.72% to 5.02%; the content of saponins in the *A. adenophora* extract ranged from 1.79% to 1.89%; the content of tannins in the *A. adenophora* extract ranged from 1.31% to 1.40%, as shown in Table 1.

## 3.3 Calculation of $LC_{50}$ of *A. adenophora* aqueous extract

The extract of *A. adenophora* was diluted to five gradient concentrations, and 50 slugs were selected for each concentration. Subsequently, the slugs were sprayed with the extract. After 7

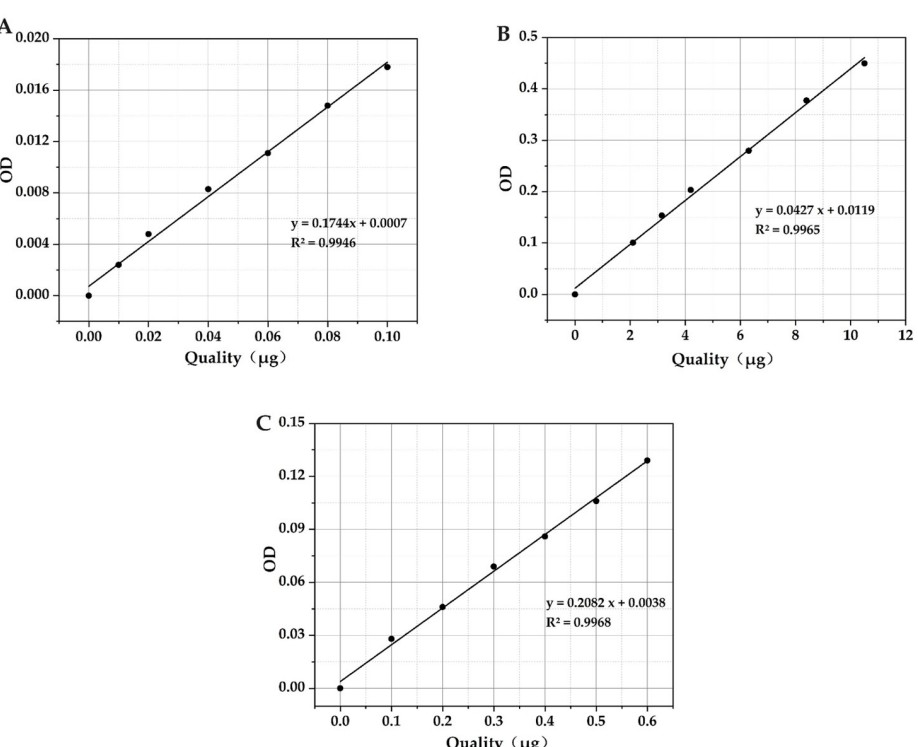

**Fig 3.** Standard curve of flavonoids, saponins and tannins (A): Rutin standard curve; (B): Ginsenoside Re standard curve; (C): Gallic acid standard curve.

**Table 1. Determination of content in *Ageratina adenophora* extract.**

| Composition | Average content (%) | Standard deviation | RSD (%) |
|---|---|---|---|
| Flavonoids | 4.87 | 0.01119 | 2.30 |
| Saponins | 1.85 | 0.041 | 2.21 |
| Tannins | 1.35 | 0.0348 | 2.57 |

days, the number of slug fatalities was recorded. The results of the toxicological test are presented in Table 2. Using statistical data from IBM SPSS Statistics 25 software and Probit for analysis to calculate $LC_{50}$, the determined $LC_{50}$ was 35.9 mg/mL. The ratio of concentration to response is shown in Fig 4.

### 3.4 Analysis of hatching rate after 24 and 48 h of egg exposure

The results demonstrated that the *A. adenophora* extract had adverse effects on the hatching rate of eggs exposed for 24 h (H = 11.3742; p = 0.0018) and 48 h (H = 20.7134; p = 0.0002). Moreover, it was observed that as the exposure time increased, the hatching rate decreased (H = 4.127; p = 0.0319). However, there was no significant difference in the average hatching value of the control group between the two periods (H = 0.0556; p = 0.5319). Specifically, the hatching rates after 24 and 48 h of exposure were 97.9% and 94.2%, respectively (Table 3).

### 3.5 Survival rate of offspring hatched from exposed eggs

In comparison with the control group, the survival rate of hatched offspring from eggs exposed to the extract experienced a significant decrease (24 h: H = 9.9698, p = 0.0037; 48 h: H = 12.9856, p = 0.0031) (Fig 5). Furthermore, the survival rate of hatched offspring from eggs exposed for 24 h was higher than that for 48 h (H = 7.5397; p = 0.0036). All slugs exposed to the extract for 48 h had perished by the 87[th] day of observation. There was no significant difference in the survival rate of hatched offspring from eggs in the control group between the two periods (24 h: 90.3%, 48 h: 87.8%; H = 7.4856, p = 0.2468) (Fig 5). Throughout the observation period, neither the slugs in the exposure group nor those in the control group displayed sexual maturity.

### 3.6 Effects of $LC_{50}$ on the growth, survival, and reproduction of newly hatched *L. maximus* larvae exposed for 24 and 48 h

Growth analysis of the exposed group revealed that at 60 days, there was no significant difference in the average size between the 24 h exposure group and the control group (H = 0.5677; p = 0.4772). However, the exposure time affected the growth rate, indicating that the longer

**Table 2. Determination of toxicity of *Ageratina adenophora* extract to adult slugs.**

| Reagent concentration (mg/mL) | Slug number | Survival number |
|---|---|---|
| 10 | 50 | 38 |
| 20 | 50 | 35 |
| 30 | 50 | 26 |
| 40 | 50 | 13 |
| 50 | 50 | 8 |

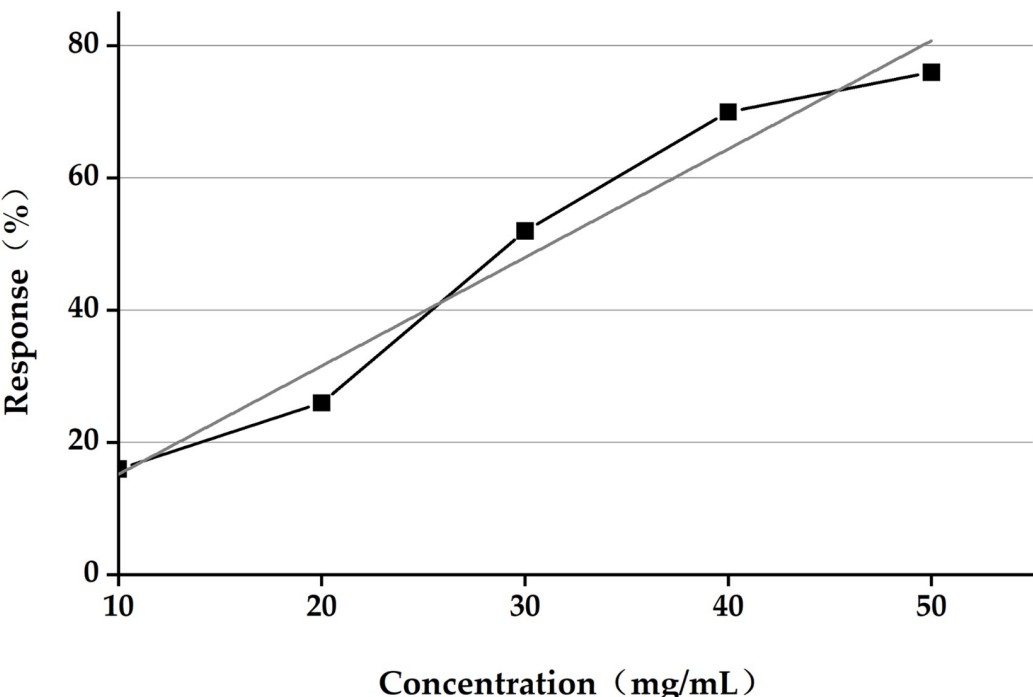

**Fig 4. Concentration–response ratio of *Limax maximus* to *Ageratina adenophora* extract.** Assessment response = fatality rate, expressed as a percentage of deceased *L. maximus*.

the exposure time, the slower the growth rate. The average size of the 48 h exposure group was lower than that of the control group (H = 8.6414; p = 0.0036). Specifically, the control group's average size at 24 and 48 h was 52.21 ± 3.94 mm and 47.66 ± 2.48 mm, respectively. In contrast, the average size in the exposed group at 24 and 48 h was 51.06 ± 3.13 mm and 36.75 ± 2.24 mm, respectively.

The survival rate of newly hatched slug larvae exposed to *A. adenophora* extract was notably lower than that of the control group (24 h: H = 5.7587; p = 0.022148h; H = 5.9039; p = 0.0221). Furthermore, a correlation was observed between the exposure time and the survival rate of slug larvae, indicating that the longer the exposure, the lower the survival rate (H = 5.0128; p = 0.0453). However, the exposure duration did not significantly affect the survival rate of the control group (H = 0.29; p = 0.5273). All slugs in the 24 h exposure group succumbed at the

**Table 3. Hatching rate of *Ageratina adenophora* aqueous extract observed after 24 and 48 h of exposure.** (Average value, standard deviation, range of variation, and hatchability percentage).

| Groups | | Hatchability X ± SD | Range of variation | Hatchability Percentage (%) |
|---|---|---|---|---|
| Control | 24 h | 39.16 ± 0.41[a] | (38–40) | 97.9 |
| | 48 h | 37.68 ± 1.09[a] | (36–39) | 94.2 |
| Exposed | 24 h | 22.64 ± 3.10[b] | (19–26) | 56.6 |
| | 48 h | 7.48 ± 1.65[c] | (5–9) | 18.7 |

[a,b,c] = means followed by distinct letters indicate significant differences as per the Kruskal–Wallis test (p < 0.05).

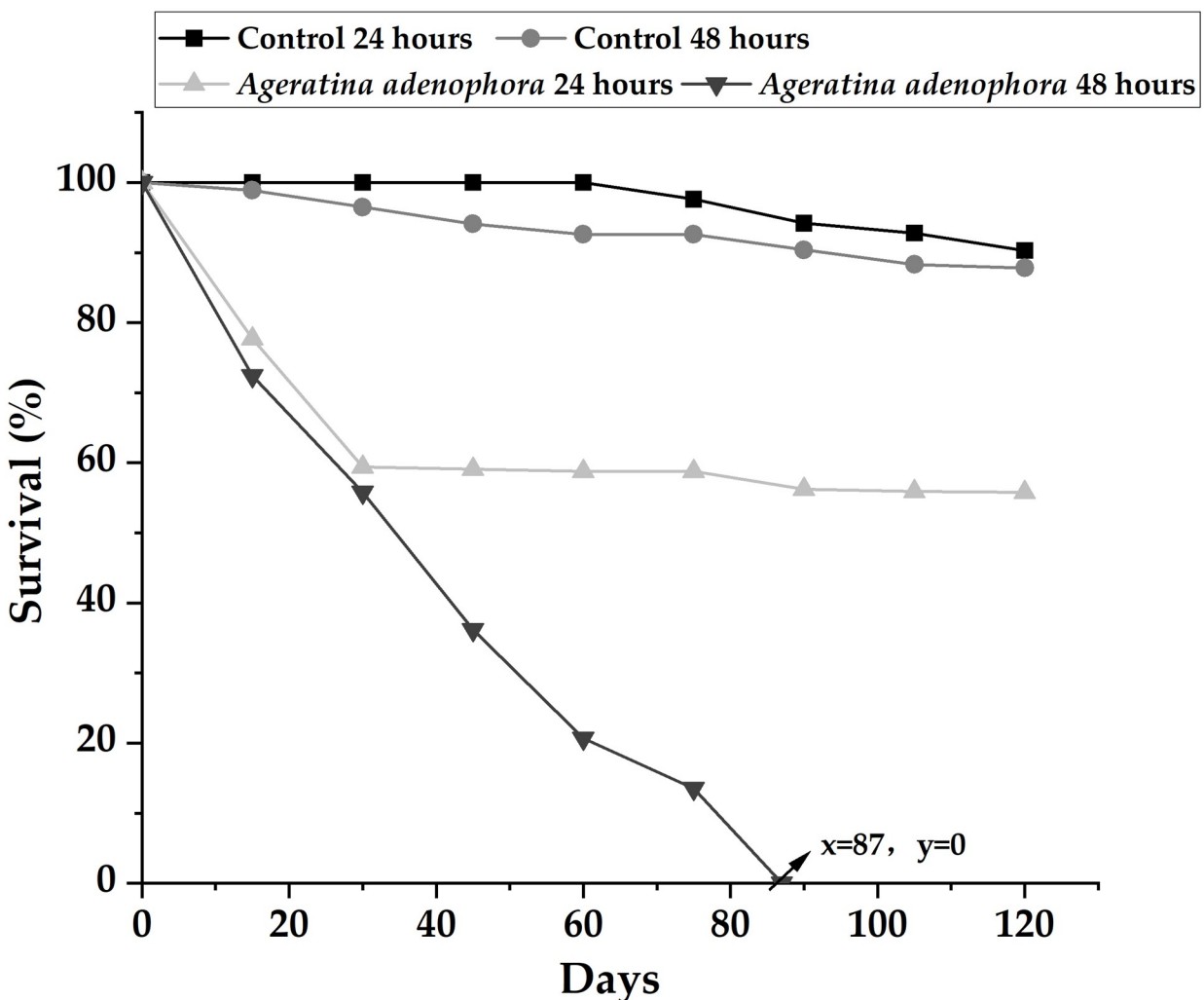

**Fig 5. The survival rate of newly hatched offspring from eggs exposed to *Ageratina adenophora* extract for 24 and 48 h over 120 days.**

end of 108 days, and those in the 48 h exposure group perished by the end of 81 days. The results are displayed in Fig 6.

At the end of the 75-day observation, the normal maturation time of *L. maximus* was consistent with the normal maturation time of the slugs. Only 28% of slugs survived in the 24 h exposure group to reach sexual maturity at approximately 68 ± 7.1 days. In comparison, the control group exhibited an average maturation time of 55 ± 4.5 days during the same period. In the 48 h exposure group, a mere 11% of slugs survived to reach sexual maturity, with an average maturation time of 73 ± 1.8 days, whereas the control group's average maturation time was 58 ± 4.9 days.

### 3.7 Effects of $LC_{50}$ on the growth, survival, and reproduction of 40-day-old *L. maximus*

Growth analysis of the exposed group revealed a higher mortality rate compared with the control group at 30 days. A significant difference in the average size of slugs between the exposed

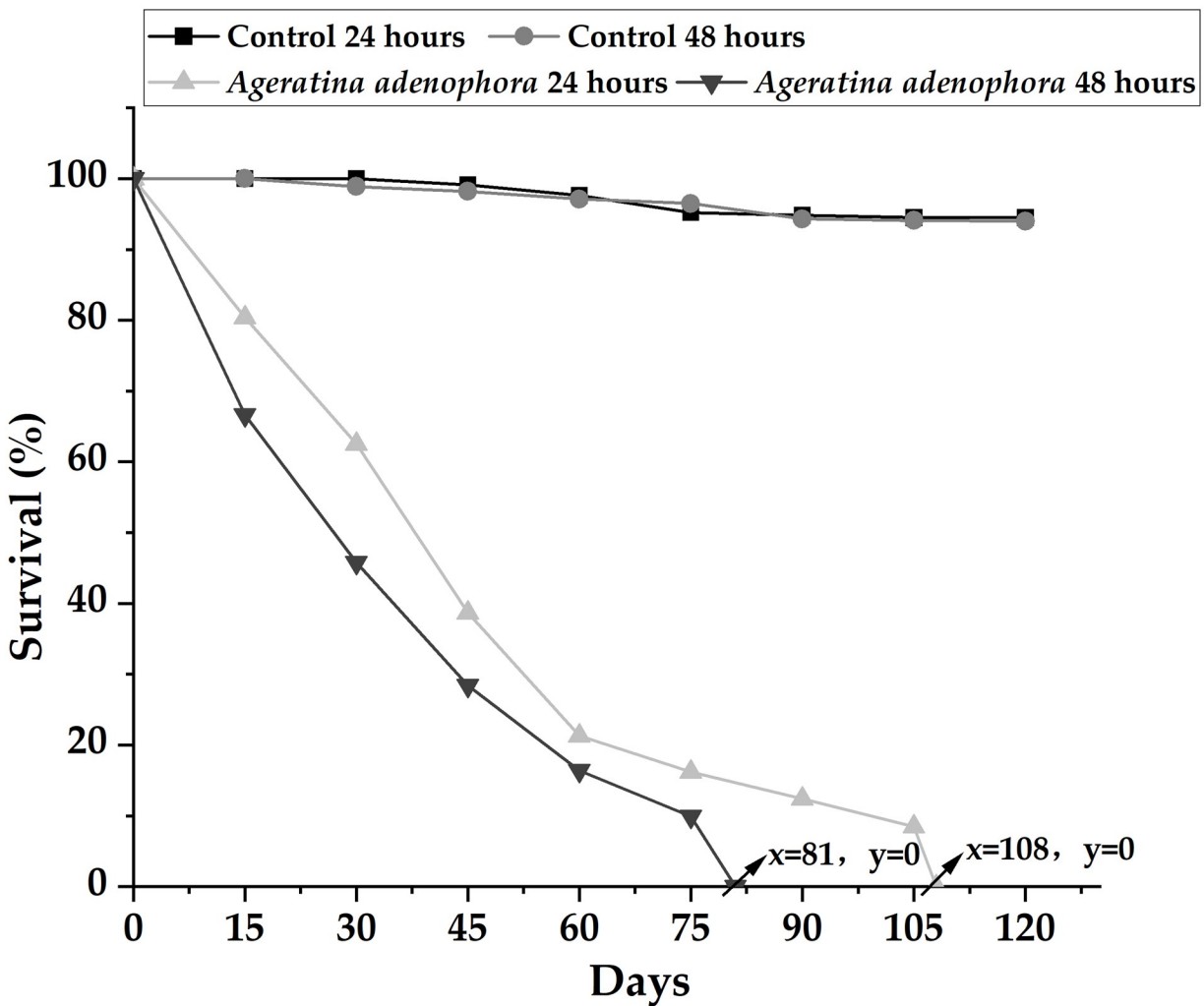

**Fig 6. Observation of newly hatched slug larvae exposed to *Ageratina adenophora* extract for 24 and 48 h over a period of 120 days.**

group and the control group was observed (24 h: H = 11.0795; p = 0.0004; 48h: H = 14.8655; p<0.0001). Specifically, the average sizes of the 24 and 48 h exposure groups were 44.96 ± 2.57 mm and 40.18 ± 2.34 mm, respectively, whereas those of the control group were 51.74 ± 3.05 mm and 50.83 ± 2.99 mm, respectively.

In the exposed group, *L. maximus* did not experience mortality on the first day, and the initial death occurred in the 48 h exposure group on the second day. All slugs in the 24 h exposure group perished at the end of 117 days, and all those in the 48 h exposure group succumbed at the end of 99 days. The survival rate of the exposure group was significantly lower than that of the control group (24 h: H = 5.4988; p = 0.0212; 48 h: H = 5.8271; p = 0.0351). Conversely, there was no significant difference in the survival rate between the two time periods in the control group (H = 0.0286; p = 0.9463.) Notably, the survival rate of slugs exposed to the extract for 48 h was higher in the first 3 weeks, then decreased gradually, and experienced a rapid decline from 30 to 75 days. These results indicate the presence of *A. adenophora* extract in *L. maximus*. The survival rate of slugs decreased linearly in the 24 h exposure group. The results are depicted in Fig 7. In the 24 h exposure group, only 19% of the surviving slugs reached

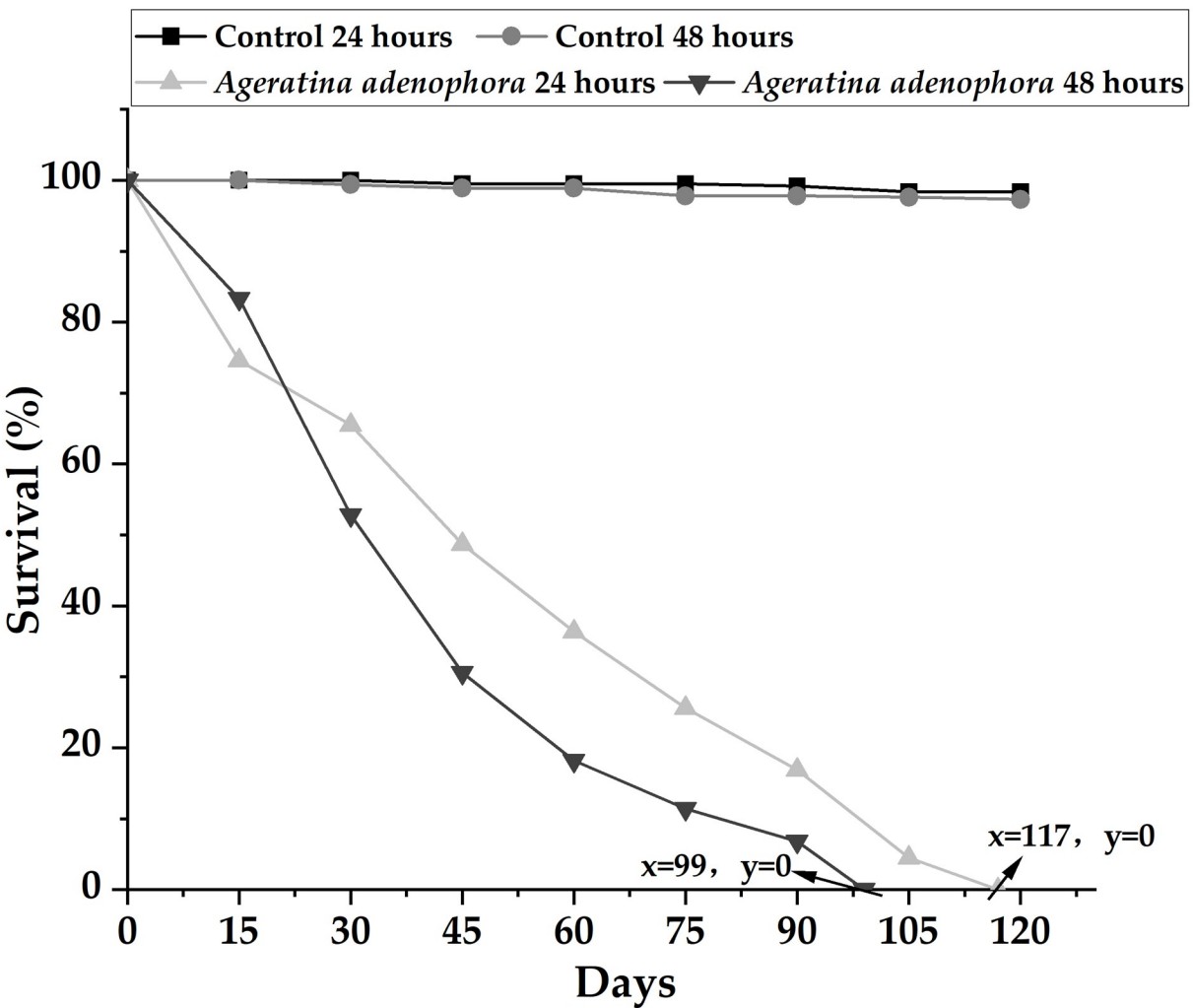

**Fig 7. Survival rate of 40-day-old *Limax maximus* exposed to *Ageratina adenophora* extract for 24 and 48 h, observed over 120 days.**

sexual maturity, with an approximate maturity period of 64 ± 5.4 days. No sexual maturity was observed in the surviving slugs in the 48 h exposure group. In contrast, the control group exhibited sexual maturity rates of 54% and 46.2% for the 24 and 48 h exposures, respectively, with average maturation times of 53 ± 3.8 days and 55 ± 2.4 days, respectively.

## 3.8 Effects of $LC_{50}$ on the activities of protective and detoxifying enzymes in newly hatched larvae and 40-day-old slugs

The effects of $LC_{50}$ on the activities of protective enzymes in slugs treated with *A. adenophora* extract for 24 and 48 h were significantly lower than those of the control (Fig 8). This finding suggests that *A. adenophora* extract can inhibit the activities of SOD, POD, and CAT enzymes. Specifically, after a 24 h treatment with *A. adenophora* extract, the activities of protective enzymes (SOD, POD, and CAT) in adult slugs decreased significantly, indicating the inhibitory effect of *A. adenophora* extract on the protective enzymes in adult slugs. This trend continued when the adult slugs were treated with *A. adenophora* extract for 48 h. The results

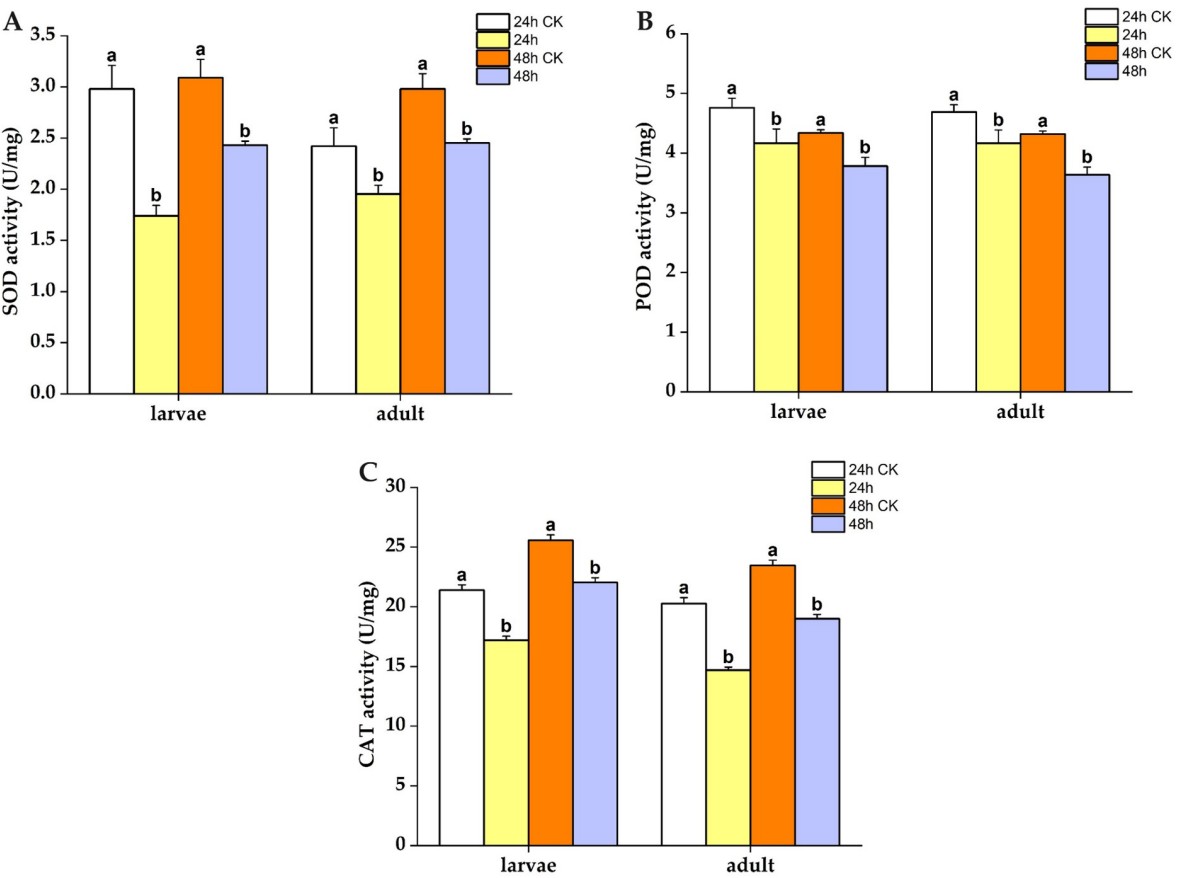

**Fig 8. Protective enzyme activities of slug larvae and adults treated with LC$_{50}$ for 24 and 48 h, respectively.** The letters above the columns denote the results of multiple comparative experiments performed using Tukey tests on the enzyme activities of slug larvae and adults after LC$_{50}$ treatment. Different lowercase letters indicate statistically significant differences. (A): SOD; (B): POD; (C): CAT.

demonstrate that the activities of protective enzymes in slugs were impacted upon treatment with *A. adenophora* extract.

The LC$_{50}$ effect on the detoxifying enzyme activity in slugs treated with *A. adenophora* extract was observed over 24 and 48 h periods (Fig 9). The activity of the detoxifying enzyme AChE in both slug larvae and adults was significantly higher than that in the control group, indicating a significant activation of AChE in these groups. However, after being treated with *A. adenophora* extract for the same durations, the activities of the detoxification enzymes CYP450 and GST in slugs were significantly lower than those in the control group. This result suggests that the CYP450 and GST enzymes in adult and larval slugs were inhibited significantly following treatment with *A. adenophora* extract.

## 4. Discussion

In this study, we observed that *A. adenophora* aqueous extracts showed insecticidal activity against slugs at various life stages. The insecticidal effects were attributed to flavonoids, saponins, and tannins, which are secondary metabolites found in the extract. However, it is important to note that other undetected secondary metabolites may have potential bactericidal

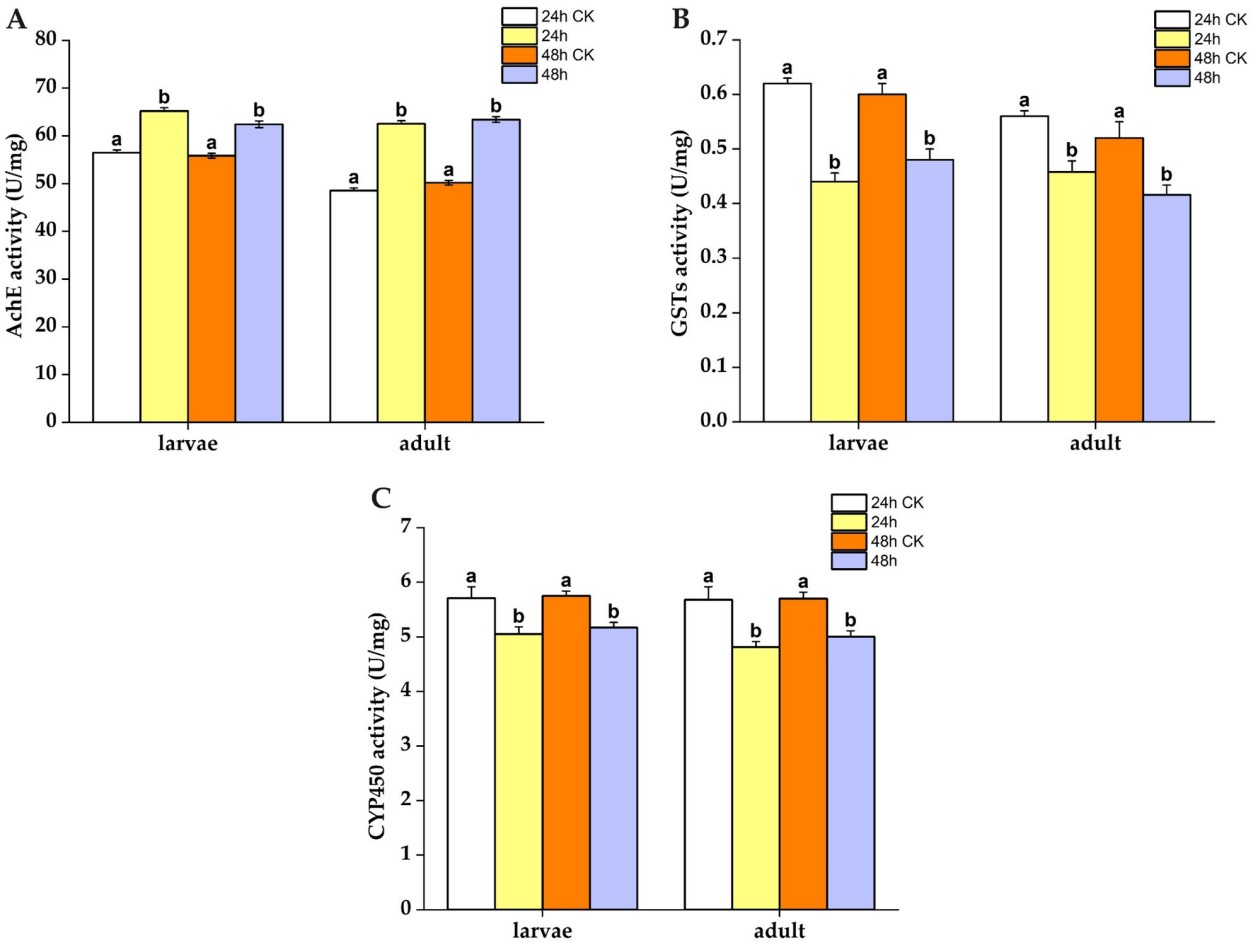

**Fig 9. Detoxifying enzyme activities of slug larvae and adults treated with LC$_{50}$ for 24 and 48 h, respectively.** The letters above the columns denote the results of multiple comparative experiments performed using Tukey tests on enzyme activities of slug larvae and adults after LC$_{50}$ treatment. Different lowercase letters indicate statistically significant differences. (A): AchE; (B): GSTs; (C): CYP450.

activity, and those with low content might still exert weak germicidal activity. Hence, further research is required to investigate the germicidal activity of these metabolites.

The *A. adenophora* extract influenced the hatching rate of *L. maximus*, indicating that the molecular structure of the active plant ingredient facilitates penetration into the embryo-associated membrane pores. However, prior research has reported that the high molecular weight of the egg membrane can impede drug penetration [26]. Notably, the mortality rate of eggs exposed to the extract for 48 h surpassed that of the 24 h exposure, signifying a more pronounced effect on the hatching rate of slug eggs with longer exposure times.

Temperature is a potential factor affecting the egg hatching rate and slug growth and development [27]. Extremes in temperature, whether high or low, can reduce mollusk vitality, leading to states of lethargy when environmental conditions are unfavorable. Additionally, high temperatures elevate mollusk oxygen consumption, accelerating the energy depletion rate within the organism [28]. In this study, the temperature was controlled within the 21°C–24°C.

Souza et al. (1992) confirmed the high sensitivity of newly hatched snails to cashew nut shell extracts [29]. In a related study, Gohar et al. (2014) reported molluscicidal activity against

newly hatched *Biomphalaria alexandrina* using *Callistemon viminalis* (Sol. ex Gaertner) G. Don ex Loudon [30]. However, our investigation demonstrated significant changes in adult *L. maximus* exposed for 24 and 48 h after 45 days, including reduced food intake, retarded movement, a darker body surface color, and somnolence. These alterations led to differences in growth, development, and sexual maturity compared with the control group [31]. Silva et al. (2020) evaluated whether the crude ethanolic extract of the *Persea americana* stem bark has molluscicidal activity against embryos and newly hatched and adult *Biomphalaria glabrata*. Results show the presence of flavonoids, anthraquinone heterosides, coumarins, and tannins in the crude ethanolic extract, which exhibited molluscicidal activity against all life cycle stages of *B. glabrata* [32]. We did not determine the existence of other secondary metabolites in the study, and cannot deny that the activity of substances with low content is low, and more substances can be determined in future studies.Under stress conditions, insects accumulate superoxide anion radicals, hydroxyl radicals, hydrogen peroxide, and other reactive oxygen species, resulting in organismal damage. However, insects possess protective enzymes, including SOD, POD, and CAT, which scavenge excess free radicals [33, 34]. These enzymes work synergistically to maintain the dynamic balance of free radical metabolism to protect the insect from injury or minimize its impact [35]. Research has indicated that *Nilaparvata lugens* and *Sogatella furcifera* significantly increased the activities of SOD, POD, and CAT in rice plants infected with rice black-streaked dwarf disease or southern rice black-streaked dwarf disease [36]. Our results implied that the $LC_{50}$ of *A. adenophora* extract had an inhibitory effect on SOD activity in slugs, inhibition–activation–inhibition effect on POD activity, and activation–inhibition effect on CAT activity.

CYP450, GST, and AChE play important roles in breaking down exogenous toxicants and maintaining normal physiological metabolism [37]. GST catalyzes sulfhydryl coupling between electrophilic groups of toxic substances and reduces glutathione, enhancing their hydrophobicity for easy excretion from the body [38]. Methylvitamin salt significantly inhibited AChE activity in the 2$^{nd}$ instar larvae of *Lymantria dispar*, and GST activity was activated after 24 h of treatment [39]. Carbendazim can inhibit AChE completely in earthworms and activate GST completely. It can induce the activation of CYP450 at low concentrations and inhibit it at high concentrations [40]. Under the sublethal concentration stress of tetrameric acetaldehyde, the activity of AChE increased initially. Then it decreased in the gill and abdominal foot of *Pomacea canaliculata*, whereas the activity in the liver and intestine decreased initially and then increased [41]. Our results demonstrated that after treatment with the sublethal concentration of *A. adenophora* ($LC_{50}$), the activities of CYP450 and GST in slugs were inhibited to varying degrees, but the activity of AChE increased. Because of the extract's high flavonoid content, AChE was released in slugs, leading to dehydration and death.

In this study, the $LC_{50}$ extract of *A. adenophora* contained the highest flavonoid content, averaging 4.87%. This was followed by saponins and tannins, which were at 1.85% and 1.35% respectively. It can be inferred that flavonoids played a significant role in the extract, although it cannot be denied that compounds in smaller quantities also exerted an effect. Future studies must confirm whether separately extracted flavonoids from *A. adenophora* extract will have a more significant lethal effect on slugs.

Flavonoids may inhibit the detoxification system of snails. Some studies have observed changes in the CYP450 enzyme of the land snail *Cantareus aspersus* after exposure to tobacco leaves [42, 43]. This study noted a decrease in CYP450 enzyme activity. after *L. maximus* came into contact with *A. adenophora* extract. This enzyme is part of a protein family involved in detoxification and can degrade various foreign substances [44]. Additionally, a study has shown that flavonoids can activate AChE, leading to a huge release of AChE in slugs [45]. This release destroys the special mucus produced by slugs, causing rapid dehydration, destruction

of body surface cells, and death due to significant body fluid loss [46]. These findings are aligned with the results of AChE enzyme activity determined in this study, supporting the idea that flavonoids can activate AChE activity *in vivo*. Previous studies have concluded that saponins exhibit hemolytic toxicity, which can destroy red blood cell membranes and cause cytoplasmic extravasation, leading to red blood cell disintegration [47, 48]. Tannin is a polyphenol containing several hydroxyl groups; it can complex with protein and cause precipitation, thereby inactivating it [49, 50]. However, the is study did not confirm the specific effects of saponins and tannins on slugs. Further research on the specific effects of these metabolites on slugs is an important future direction.

Exposure to toxic substances or gases can cause physiological stress in slugs, reducing carbohydrate and glycogen reserves [51]. This triggers the depletion of the body's protein as an alternative energy source [52]. Studies have shown that mollusks exposed to toxins exhibit increased levels of hemolymph protein and uric acid, indicating protein degradation for energy [51]. However, proteins may precipitate after complexation and cannot be used as an energy substance [53], resulting in an energy imbalance and ultimately death in the slug.

Currently, chemical agents are predominantly used for mollusk control in Chinese agriculture [3]. However, the drawbacks of chemical control are becoming increasingly apparent. Among them, the chemical 3R problem has become a recognized and urgent problem to be solved all worldwide. 3R problems include that organisms have been treated with pesticides for a long time to make their offspring resistant to drugs and the pesticides control other pests and kill the natural enemies of the population and the residues of chemical pesticides on agricultural products [54]. Plant extracts should be effective against all life stages of the target organism for more effective control of these species. Considerations for agricultural application include factory production; extract preparation and treatment; selectivity at low concentrations; and potential harm to humans, the environment, and crops. Comprehensive studies on plant secondary metabolites in terrestrial slug species should be strengthened. This study confirmed that *A. adenophora* extract has a lethal effect on slugs, but it is unclear which secondary metabolites have the greatest killing activity. Future research should focus on a specific class of secondary metabolites or combinations to maximize the lethal effect. As an invasive plant, *A. adenophora* is abundant and can be extracted via simple water extraction processes, which reduces the cost and technical difficulty for production in the factory [55]. Our extract still exhibits bactericidal activity after being stored in refrigerator at 4°C for 6 months, imparting the extract with a long service life. Moreover, field experiments are necessary for practical farmland control applications.

## 5. Conclusions

*A. adenophora* is an invasive plant characterized by its wide distribution and strong fecundity. The aqueous extract of *A. adenophora*, which contains flavonoids, saponins, and tannins, has been shown to affect the activity of *L. maximus* at all stages of the yellow slug's life cycle. This extract inhibits egg hatching and induces morphological changes in eggs. Interestingly, adults appear to be more sensitive to the extract's toxicity than larvae. In addition to significant body fluid loss, behavioral changes, such as lethargy and decreased vitality, were observed in adult slugs. These findings suggest that the extract disrupts the balance within *L. maximus*. Therefore, the aqueous extract of *A. adenophora* contains active ingredients that can kill slugs. Given its water solubility, the extraction process will likely be environmentally friendly. Furthermore, it holds promise to effectively control the biological invasion of *A. adenophora* and slug species that threaten other agriculturally important crops.

## Supporting information

**S1 Fig. The growth of *L. maximus* was compared with the control 24 h after 30 days of treatment.** (A):Control; (B):Exposed; (C):Compare.
(TIF)

## Acknowledgments

The authors wish to thank the Institute of Entomology of Guizhou University for assistance in this research.

## Author Contributions

**Conceptualization:** Haojun Li, Wenlong Chen.

**Formal analysis:** Runa Zhao, Yingna Pan.

**Investigation:** Haojun Li, Yingna Pan.

**Methodology:** Haojun Li, Hui Tian.

**Validation:** Runa Zhao.

**Writing – original draft:** Haojun Li.

**Writing – review & editing:** Runa Zhao, Wenlong Chen.

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
