## [Decision Letter · Decision Letter 0]

13 Nov 2023

PONE-D-23-33974Insecticidal activity of Ageratina adenophora (Asteraceae) extract against Limax maximus (Mollusca, Limacidae) at different developmental stages and its chemical constituent analysisPLOS ONE

Dear Dr. li,

Thank you for submitting your manuscript to PLOS ONE. After careful consideration, we feel that it has merit but does not fully meet PLOS ONE’s publication criteria as it currently stands. Therefore, we invite you to submit a revised version of the manuscript that addresses the points raised during the review process.

We look forward to receiving your revised manuscript.

Kind regards,

Vivekanandhan Perumal, Ph.D

Academic Editor

PLOS ONE

Journal Requirements:

Additional Editor Comments:

Comments to the authors

The manuscript title is “Insecticidal activity of Ageratina adenophora (Asteraceae) extract against Limax maximus (Mollusca, Limacidae) at different developmental stages and its chemical constituent analysis” by Li et al. After carefully reviewing, I can see some merit in this research. The current form is not up to the scientific level, so please carefully fix the following errors.

The introduction part is poorly written, and several details are missing in the current form. Please add information about current control methods and their disadvantages, then write about alternative biological control methods. Only then will the reader understand clearly; otherwise, they will be confused.

Line 60: typo error

Line 66: Why did you dry plant materials at 60°C for 24 hours? because at 60 oC, most of the volatile compounds will degrade, so please check it.

Line 66: Please add the made and made of the mechanical grinder.

Why have you selected only distilled water for extraction? What about other solvents?

Figures 1–9: Please change them and add high-quality images.

Reviewers' comments:

Reviewer's Responses to Questions

**Comments to the Author**

1. Is the manuscript technically sound, and do the data support the conclusions?

Reviewer #1: Yes

Reviewer #2: Yes

2. Has the statistical analysis been performed appropriately and rigorously? 

Reviewer #1: Yes

Reviewer #2: Yes

3. Have the authors made all data underlying the findings in their manuscript fully available?

Reviewer #1: No

Reviewer #2: Yes

4. Is the manuscript presented in an intelligible fashion and written in standard English?

Reviewer #1: Yes

Reviewer #2: Yes

5. Review Comments to the Author

Reviewer #1: 1. Good manuscript but can be improved further.

2.Improve on your data presentation to be more concise, and include more interesting points in your discussions.

3. Language can be improved as well.

4. Please see more on the comment attachment.

Reviewer #2: The manuscript Insecticidal activity of Ageratina adenophora (Asteraceae) extract against Limax maximus (Mollusca, Limacidae) at different developmental stages and its chemical constituent analysis, has been written well but needs minor revisions.

6. PLOS authors have the option to publish the peer review history of their article (what does this mean?). If published, this will include your full peer review and any attached files.

Reviewer #1: **Yes: **Rajiv Ravi

Reviewer #2: No

---

## [Author Response · Author response to Decision Letter 0]

4 Jan 2024

Dear Editor and Reviewers,

Thanks very much for taking your time to review this manuscript. I really appreciate all your comments and suggestions! Please find my itemized responses in below and my revisions/corrections in the re-submitted files

SUGGESTIONS FROM EDITOR

The introduction part is poorly written, and several details are missing in the current form. Please add information about current control methods and their disadvantages, then write about alternative biological control methods. Only then will the reader understand clearly; otherwise, they will be confused.

Line 60: typo error

Line 66: Why did you dry plant materials at 60°C for 24 hours? because at 60 oC, most of the volatile compounds will degrade, so please check it.

Line 66: Please add the made and made of the mechanical grinder.

Why have you selected only distilled water for extraction? What about other solvents?

Figures 1–9: Please change them and add high-quality images.

Academic Editor:

1. I have perfected the introduction of the manuscript, adding the current control methods and shortcomings, and explaining the alternative biological control methods, and why plant extracts are selected for control.

2. Line 60: typo error (Modified)

3. Line 66: Why did you dry plant materials at 60°C for 24 hours?

I chose to dry at 60℃ for 24 hours because I referred to someone else's method. Most plant extracts need to be dried before they are prepared. Plant extracts mainly play a role in secondary metabolites, which can only be extracted by heating or even evaporation, and the extracts in this study still have obvious toxic effect on Limax maximus.

4. Line 66: Please add the made and made of the mechanical grinder. (Added)

5. Why have you selected only distilled water for extraction? What about other solvents?

First of all, distilled water is selected because its influencing factors are small and suitable as a control, and pre-experiments have been done to confirm that it has no toxic effect on Limax maximus, and other solvents may have a better effect. This is a point that needs to be improved in our research in the future. Thank you very much for your suggestion.

6. Figures 1–9: Please change them and add high-quality images. (Changed)

Reviewer #1:

Abstract

1. Please include the LC50 value specifically for 24 hours and 48 hours

The LC50 value in our study was calculated by virulence test. 24 h and 48 h were the treatment time set by the extract with LC50 concentration, so there was only one LC50 value.

2. In my opinion, there should be two values of the LC dose, might be lower for longer exposure time

LC50 is a semi-lethal concentration, the specific meaning is that in the acute toxicity test of animals, the concentration of poison that makes half of the animals killed is generally only one value, and the virulence test is carried out at the same time, the difference is that the concentration of dilution is different.

Introduction

1. Expand and explain line 45-48 more the use of this plant to control mollusk species

At present, the example of using Ageratina adenophora to control molluscs is still very rare. the purpose of using Ageratina adenophora extract on Limax maximus in our study is to explore whether this extract has killing effect on Limax maximus. in order to develop a new botanical pesticide.

2. What other studies have used this plant for controlling the mollusk species. What is the specific chemical component novel from this plant which has been used for biocidal activity?

No other studies have used this plant to control molluscs, and the special chemical components we have identified are flavonoids, saponins and tannins, all of which have bactericidal activity. however, we do not rule out the fact that other undetected chemical components also have killing activity, which is also a direction of our future research.

Materials and Methods

1. Line 56-69, please include pictures for slug and plants with its morphological identification about their species. Verification of species information required for both slug and plant

We have added pictures of Limax maximus and Ageratina adenophora to the manuscript, because we have given detailed species names, and this study does not belong to the taxonomic category, and the corresponding species can be found well according to the species names.

Results

1. Please include a summary table for flavonoids, saponins and tannins, to enable readers for quick look at the content compositions.

We have attached a summary table to make the data more intuitive.

2. Please include some interesting pictures showing the growth analysis, size differences between control and treatment. Line 280-286 etc

We added a growth comparison chart to the Supporting Information Captions.

Discussions

1. Line 382-387, add more discussions on previous studies using plant extracts in relation to molluscicidal activity

We added some discussions about the use of plant extracts to control molluscs.

2. Discuss the advantage of your extracts compare to chemical usage. Please include some literatures about the degradation profile on plant extracts compared to chemical treatments. Can your natural plant extracts be degraded fast in actual environment compared to chemical usage?? 

Chemical agents are often accompanied by 3R problems, and there are few articles about the degradation of plant extracts compared with chemicals. At present, the use of plant extracts as a new chemical is still in the experimental stage. Field experiments and degradation analysis have not yet been carried out. In order to really apply it to practical control, we will speed up our research to ensure compliance with the principles of the use of pesticides. Thank you for your suggestions.

3. Explain more line 436-445 on chemicals harm to human and environment. What is your plant extracts applicability in industrial approach.

We have explained the harm of chemicals to human beings and the environment, as well as the convenience of our extracts in factory production.

4. What can be the degradation profile, shelf life for your extracts, how long can your extract will be effective. Example, can your extract still be effective after keeping in solution preps for more than 1-3 months.

The degradation of our extract has not been studied yet, which is an important indicator for us to apply it to field experiments. Our extract can still be effective after 1-3 months of preservation, which has been explained in the manuscript.

Reviewer #2:

We modified some sentence patterns and phrases according to the requirements of the reviewer#2.

We would like also to thank you for allowing us to resubmit a revised copy of the manuscript.

We hope that the revised manuscript is accepted for publication in the Plos One.

---

## [Decision Letter · Decision Letter 1]

25 Jan 2024

PONE-D-23-33974R1Insecticidal activity of Ageratina adenophora (Asteraceae) extract against Limax maximus (Mollusca, Limacidae) at different developmental stages and its chemical constituent analysisPLOS ONE

Dear Dr. li,

Thank you for submitting your manuscript to PLOS ONE. After careful consideration, we feel that it has merit but does not fully meet PLOS ONE’s publication criteria as it currently stands. Therefore, we invite you to submit a revised version of the manuscript that addresses the points raised during the review process.

We look forward to receiving your revised manuscript.

Kind regards,

Vivekanandhan Perumal, Ph.D

Academic Editor

PLOS ONE

Journal Requirements:

Additional Editor Comments:

Editor comments

After revising the manuscript, it now meets scientific standards. However, the quality of the images is poor. I kindly request the authors to provide high-quality images for Figures 1 and need scale bar, 3, 4, 5, 6, and 7. The high quality of images helps strengthen your paper. I hope you understand.

Reviewers' comments:

Reviewer's Responses to Questions

**Comments to the Author**

1. If the authors have adequately addressed your comments raised in a previous round of review and you feel that this manuscript is now acceptable for publication, you may indicate that here to bypass the “Comments to the Author” section, enter your conflict of interest statement in the “Confidential to Editor” section, and submit your "Accept" recommendation.

Reviewer #1: All comments have been addressed

Reviewer #2: All comments have been addressed

2. Is the manuscript technically sound, and do the data support the conclusions?

Reviewer #1: Yes

Reviewer #2: Yes

3. Has the statistical analysis been performed appropriately and rigorously? 

Reviewer #1: Yes

Reviewer #2: Yes

4. Have the authors made all data underlying the findings in their manuscript fully available?

Reviewer #1: Yes

Reviewer #2: (No Response)

5. Is the manuscript presented in an intelligible fashion and written in standard English?

Reviewer #1: Yes

Reviewer #2: Yes

6. Review Comments to the Author

Reviewer #1: Comments has been accepted and changes is significantly good for acceptance. The manuscript is good to be accepted as, it shows a preliminary findings on the plant extracts and potentially can be novel discovery for plant extracts. Furthermore, the graphs and data sets supports the findings.

Reviewer #2: The manuscript Insecticidal activity of Ageratina adenophora (Asteraceae) extract against Limax maximus (Mollusca, Limacidae) at different developmental stages and its chemical constituent analysis has been modified well, now it may be Accepted for publication in plos one

7. PLOS authors have the option to publish the peer review history of their article (what does this mean?). If published, this will include your full peer review and any attached files.

Reviewer #1: No

Reviewer #2: No

---

## [Author Response · Author response to Decision Letter 1]

26 Jan 2024

Dear Additional Editor,

Thanks very much for taking your time to review this manuscript. I really appreciate all your comments and suggestions! Please find my itemized responses in below and my revisions/corrections in the re-submitted files

SUGGESTIONS FROM EDITOR

We have uploaded our figure files to the Preflight Analysis and Conversion Engine (PACE) digital diagnostic tool to ensure that our pictures meet the requirements. We added the scale bar to Fig 1 and the clarity of Fig 3, 4, 5, 6 and 7 to make it look clearer.

Finally, we checked our reference list to make sure that our reference list were complete and accurate and not rejected.

We would like also to thank you for allowing us to resubmit a revised copy of the manuscript.

We hope that the revised manuscript is accepted for publication in the Plos One.

---

## [Editor Report · Decision Letter 2]

30 Jan 2024

Insecticidal activity of Ageratina adenophora (Asteraceae) extract against Limax maximus (Mollusca, Limacidae) at different developmental stages and its chemical constituent analysis

PONE-D-23-33974R2

Dear Dr. li,

We’re pleased to inform you that your manuscript has been judged scientifically suitable for publication and will be formally accepted for publication once it meets all outstanding technical requirements.

Kind regards,

Vivekanandhan Perumal, Ph.D

Academic Editor

PLOS ONE

Additional Editor Comments (optional):

The authors have responded to my comments, and now the manuscript meets scientific standards. Therefore, I recommend this manuscript for publication.
---

## [Editor Report · Acceptance letter]

14 Feb 2024

PONE-D-23-33974R2 

PLOS ONE

Dear Dr. Li, 

I'm pleased to inform you that your manuscript has been deemed suitable for publication in PLOS ONE. Congratulations! Your manuscript is now being handed over to our production team.

Kind regards, 

on behalf of

Dr. Vivekanandhan Perumal 

Academic Editor

PLOS ONE